# Pooled prevalence of stunting and associated factors among children aged 6–59 months in Sub-Saharan Africa countries: A Bayesian multilevel approach

**Bayley Adane Takele** *, **Lemma Derseh Gezie, Tesfa Sewunet Alamneh**

Department of Epidemiology and Biostatistics, Institute of Public Health, College of Medicine and Health Sciences, University of Gondar, Gondar, Ethiopia

* behaileadane@gmail.com

## Abstract

### Background

Over 155 million children under five suffer from stunting, and it is responsible for over one million deaths and 54.9 million Disability Adjusted Life Years (DALYS) of under-five children worldwide. These predominantly occurred in low-and middle-income countries like sub-Saharan Africa. Stunted children begin their lives at a marked disadvantage. Some of these are; poor cognition and educational performance, low adult wages, lost productivity and, when accompanied by excessive weight gain later in childhood, an increased risk of nutrition-related chronic diseases in adult life and the devastating effects of stunting can last a lifetime and even affect the next generation. Despite this, its magnitude rises in the past 25 years in sub-Saharan Africa. Studies that capture the pooled prevalence and associated factors of stunting among children aged 6–59 months in sub-Saharan Africa are limited. Therefore, this study was carried out on the basis of the Bayesian approach to determine the pooled prevalence and predictors of stunting among children aged 6–59 months in Sub-Saharan Africa.

### Objective

To assess the pooled prevalence of stunting and associated factors among children aged 6–59 months in Sub-Saharan Africa

### Methods

For this study a total of 173,483 weighted samples from the demography and health survey data set of 35 sub-Saharan African countries from 2008 to 2020 were used. After checking Variation between cluster by computing Intraclass Correlation Coefficient, binary logistic regression model was conducted based on hierarchical Bayesian statistical approach to account the hierarchical nature of demography and health survey data and to get reliable estimates by using additional information from the prior distribution. Adjusted odds ratio with 95% credible interval of the best fitted model was used to ascertain the predictors.

**Data Availability Statement:** Data is available online and it can be accessed from www.measuredhs.com.

**Funding:** The author(s) received no specific funding for this work.

**Competing interests:** The authors have declared that no competing interests exist.

**Abbreviations:** AOR, Adjusted Odds Ratio; CI, Confidence Interval; CrI, credible interval; DALYS, Disability Adjusted Life Years; DHS, Demographic Health Survey; EA, Enumeration Area; ICC, Intra-cluster Correlation Coefficient; LOOIC, Leave-One-Out Information Criteria; MOR, Median Odds Ratio; PCV, Proportional Change in Variance; SD, Standard Deviation; SSA, Sub-Saharan Africa; UNICEF, United Nations International Children Fund; WHO, World Health Organization.

## Results

The pooled prevalence of stunting in Sub-Saharan Africa was about 35% (95%CI: 34.87, 35.31). Of the sub-regions, the highest prevalence of stunting was in East Africa, 37% (95%, CI: 36.96, 37.63) followed by Central Africa, 35% (95%CI: (34.93, 35.94). Being male (AOR = 1.27, 95% CrI 1.25, 1.30), small birth size (AOR = 1.29, CrI 1.25, 1.32), home delivery (AOR = 1.17, CrI 1.14, 1.20), and no education of mothers (AOR = 3.07, CrI 2.79, 3.39) were some of the significant predictors of stunting of children.

## Conclusion and recommendation

The prevalence of stunting of children in sub-Saharan Africa is among the highest in the world. Predictors such as being male, being small at birth, a child delivered at home, and, low level of maternal education were some of the predictors of childhood stunting. Stakeholders and non-governmental organizations should consider those contributing factors of stunting when they plan and design nutritional improvement programs.

## Background

A child is considered to be stunted when their height for age is less than minus two standard deviations below the World Health Organization (WHO) child growth standard [1]. Stunted children are inches shorter than they would be under healthier circumstances, and can appear years younger. Stunting is the devastating result of poor nutrition in early childhood. Stunted children begin their lives at a marked disadvantage. Some of these are; poor cognition and educational performance, low adult wages, lost productivity and, when accompanied by excessive weight gain later in childhood, an increased risk of nutrition-related chronic diseases in adult life and the devastating effects of stunting can last a lifetime and even affect the next generation [1].

Children experience stunting from poor nutrition, repeated infection, and inadequate psychosocial stimulation. Although problems related to poor nutrition affect the entire population, children are more vulnerable because of their unique physiology and socioeconomic characteristics [2].

Over 155 million children under five suffer from stunting [3, 4], and it is responsible for over one million deaths and 54.9 million Disability Adjusted Life Years (DALYS) of under-five children worldwide. These predominantly occurred in low-and middle-income countries like SSA.

According to the WHO and UNICEF joint report the Global prevalence of stunting was 23.1% in 2015 while in Africa it was 30.9% in 2015 [5]. Evidence from thirty-five low and middle-income countries revealed that the prevalence of stunting was 38.5% [6].

Stunting is a persistent problem for young children in sub-Saharan Africa. A high percentage of malnourished children fail to reach the normal international standard of height for their age, i.e., they are stunted [7]. Contrary to global trends where the number of stunted children have been declining over the last 25 years, in Eastern and Southern Africa the number of stunted children has risen from 23.6 million to 26.8 million in the same period, due to slow rates of stunting reduction and a quickly expanding child population [8].

Stunting can result from several causes but in developing countries like SSA nutritional deficiencies, infections and poor sanitations are the major contributing factors of stunting.

Previous studies revealed that socio-demographic, economic and health related factors are significantly associated with stunting status of children. These includes sex of the child [9–12], small birth size [10, 13, 14], birth type (twine) [15, 16], breastfeeding [17, 18], history of diarrhea [19–22], history of fever, cough, unimproved water source and unimproved toilet facility [23, 24], maternal education [25, 26], poor wealth index [6, 27, 28], place of residence [29, 30], short birth interval [31, 32], mass media exposure [28, 33] and home delivery [34]).

Although there are pocket studies conducted in different sub-Saharan African countries on prevalence of stunting and associated factors [12, 14, 35, 36], as far as our search of literature studies that capture the pooled prevalence and associated factors of stunting among children aged 6–59 months are limited. Conducting pooled analysis has many advantages like it increases the sample size that determines the power of the study to detect the true effect size and to enhance the generalizability of the study findings as well as ability to compare outcome between sub-regions. Additionally, conducting pooled analysis has advantages in the public health perspective where sub-Saharan African countries are the hotspot areas of childhood undernutrition, and therefore, identifying common factors associated with stunting could help international, regional and sub-regional organizations to design public health programs at regional and subregional level which intern can enable countries to share good stunting reduction experiences, and to cooperate and pool their potential in combating stunting of children. Contradictory factors affecting stunting of children were reported in previous studies. For example, some studies had been reported that being a male child was associated with increased stunting [9, 37, 38] but a study conducted in Ethiopia showed that female children were associated with increased stunting [12].

Therefore, the aim of this study was to assess the pooled prevalence of stunting and associated factors among under-five children in sub-Saharan Africa using a multilevel binary logistic Bayesian approach.

## Methods and materials

### Data source and sampling technique

A secondary data analysis was conducted based on 35 SSA countries' DHS data set from 2008 to 2020. SSA has four sub-regions namely; Central Africa, East Africa, West Africa and Southern Africa. According to global trends in 2020, the population of SSA was 1.1 billion [39]. In 2015, Sub-Saharan Africa was home to 27 of the world's 28 poorest countries and had more extremely poor people than in the rest of the world combined [40]. Recent DHS data set of 35 SSA countries were appended together to investigate the pooled prevalence and determinants of stunting among children aged 6–59 months in SSA. The DHSs were a nationally representative survey that collected data on basic health indicators like mortality, morbidity, family planning service utilization, fertility, maternal and child health-related indicators including height and weight of under-five children. The data was derived from the measure DHS program, and the detailed information about the surveys can be found in each country' DHS report.

Demography and health survey sample designs are usually two-stage stratified probability samples drawn from an existing sample frame, generally the most recent census frame. Typically, DHS samples are stratified by geographic region and by urban/rural areas within each region. Within each stratum, the sample design specifies an allocation of households to be selected. In the first stage of selection, the primary sampling units (PSUs) are selected with probability proportional to size (PPS) within each stratum. The PSU forms the survey cluster. In the second stage, a complete household listing is conducted in each of the selected clusters. Following the listing of the households, a fixed number of households is selected by equal probability systematic sampling in the selected cluster. For this study, we used the Kids Record

dataset (KR file). Using the KR file, we extract the dependent and independent variables for each country and then we appended, and pooled the DHS survey data of the 35 sub-Saharan Africa countries.

Standard DHS Surveys have large sample sizes (usually between 5,000 and 30,000 households) and typically are conducted about every 5 years, to allow comparisons over time. For this study, a total weighted sample of 173,483 of under-five children aged 6–59 months from the DHS data set of 35 sub-Saharan African countries were used S1 Table in S1 File.

## Study variables and measurements

**Dependent variable.** Stunting status of under-five children: height for age of children dichotomized as normal (not stunted) if height for age $\geq$ -2 SD and stunted if height of age < -2 SD form WHO child growth reference. Weight measurements were obtained using light-weight SECA mother-infant scales with a digital screen designed and manufactured under the guidance of UNICEF. Height measurements were obtained using the Shorr measuring board. Children younger than 24-months were measured for their height while lying down, and children older than 24 months were measured while standing.

**Independent variables.** In line with the study's objectives and due to the hierarchical nature of the DHS data where children and mothers were nested within a cluster, two level independent variables were considered.

**Community-level or second level variables**: sub-regions of SSA, residence, distance to a health facility.

**Individual-level variables.** Categorized as socio demographic and economic factors including; child's age, child's sex, breastfeeding (ever breastfeed an currently breastfeeding), maternal age, marital status, maternal educational level, wealth index, media exposure, sex of household head, type of water source, type of toilet facility, and obstetrical and/ health-related factors include; place of delivery, a recent history of diarrhea, recent history of cough, deworming of child, recent history of fever, immunization, type of birth, size of the baby at birth and subsequent birth interval.

## Data management and analysis

A secondary data analysis was done based on the most recent DHS datasets of 35 SSA countries from 2008 to 2020. First, we have requested the DHS measure program permission for the data access by explaining the purpose of the study. Once accessed and downloaded the data set, we have appended the 35 countries' DHS data set in to one dataset using the Stata command "append using". Then we have developed a structured checklist based on literature. After that we have kept plausible variables using our checklist. Then after we had removed missing records in the dependent variable, and categorized the outcome, height for age, as normal and stunted based on the WHO growth reference. We had also categorized, recategorized and recoded our independent variables based on literature and according to DHS guide 7. Furthermore, we have made all the data management process using Stata version 14 and the data was weighted using sampling weight, primary sampling unit, and strata before any statistical analysis done to restore the representativeness of the survey.

We had conducted meta-analysis, and Stata version 14 software was used to compute the pooled prevalence and construct the forest plot [41]. Since there was heterogeneity between countries' surveys, we had computed sub-group prevalence using random effect methods, and the variance estimation method used was Restricted Maximum Likelihood (REML). Finally, the data was exported to R version 4.1.0 statistical software [42], and we used the Bayesian Regression Model Using Stan (BRMS) of R package to conduct the binary logistic regression

analysis based on Bayesian hierarchical statistical approach to identify associated factors of stunting [43]. Results were presented in the form of texts, tables and figures.

## Bayesian multi-level binary logistic regression model

Since the outcome variable is dichotomized (normal and stunted), a binary logistic regression model was employed.

Given the DHS data has hierarchical nature, participants are nested within a cluster, and it is natural to assume that study subjects in the same cluster may share similar characteristics to participants in another cluster. This violates the independence observations and equal variance assumptions between clusters of the logistic regression model. This implies the need to take into account the heterogeneity between clusters by using an appropriate model. The data has two levels with a group of J EAs and within-group j (j = 1, 2. . ., J), a random sample nj of level-one units (individual children). The response variable is denoted by;

$$Y_{ij} = 0, \text{ if the } i^{th} \text{ children are in the } j^{th} \text{ EAs had no stunting}$$

$$1, \text{ if } i^{th} \text{ children are in the } j^{th} \text{ EAs had stunting}$$

Therefore, a multilevel binary logistic model was fitted to get a reliable estimate and standard error in such data that has nested nature so as to make appropriate inferences and conclusions [44].

Four models were constructed for the multilevel binary logistic regression analysis. The first model was an empty model without any explanatory variables, to calculate the extent of cluster variation on stunting status. Variation between clusters (EAs) was assessed by computing Intraclass Correlation Coefficient (ICC), Median Odds Ratio (MOR) and proportional change in variance (PCV) (the equations and interpretations of these entities can be found in the supplementary material).

The second model was fitted with individual level variables; the third model was adjusted for community level variables while the fourth was fitted by incorporating both individual and community level variables together.

The statistical analysis was conducted based on the Bayesian statistical approach which considers the population parameters as unknown and random entities that follow a certain probability distribution. This approach has three components which include the likelihood function, prior distribution, and posterior distribution [45].

**Likelihood function.** The likelihood function is one component of the Bayesian statistical approach that exploits information about the parameters contained in the data at hand. In our case, the data to be used has a binary response with two hierarchies which needs to employ multilevel binary logistic regression analysis.

**Prior distribution.** It is the probability distribution that expresses prior uncertainty associated with the parameter of interest. There are two common types of priors in Bayesian statistics (Informative and Non-informative priors). An informative prior is a prior distribution that is used when information about the parameter of interest is available before the data is collected and the information can be obtained from previous studies, expert knowledge or experience and a combination of both but prior information should not be obtained from the currently collected data. On the other hand, when there is not enough prior information non informative (uniform or flat) prior, which gives less weight to the prior knowledge can be used. For this study, non-informative priors; Normal flat (0, 1000) prior distribution for the fixed (population and intercept) parameters and uniform (0, 1000) prior distribution for random effect (group level) were used.

**Posterior distribution.** The posterior distribution is a method to summarize what we know about uncertain quantities after the data has been observed in the Bayesian statistical analysis. It can be obtained by multiplying the prior distribution normal for β parameters and the binary logistic regression likelihood function. The posterior distribution is derived from the well-known Bayes' theorem (the formulation is supplemented).

Since Bayesian estimation needs complex calculation, approximation using simulation is required. For this study, Simulation was applied by using a BRMS package with two chains each of 3000 with 1000 warm up iterations, and all the parameters were left to initiate randomly. In our case, a variant of Markov chain Monte Carlo (MCMC) methods called No-U turn sampler (NUTS) was used. NUTS sampler improves the drawback of Hamiltonian Monte Carlo's (HMC) by introducing the slice variable sampled uniform distribution in the sampling procedure.

The results from a given distribution are not deemed reliable until the chain has reached its stationary assumption. But the inference becomes appropriate when target distributions become well converged. Therefore, monitoring its convergence is essential for producing reliable results from the posterior distribution. The convergence of the targeted distribution was assessed by trace plot, density plot, effective sample size and R hat statistics.

Model selection criteria such as Leave-one-out cross-validation (LOO) and Watanabe Akaike information criterion (WAIC) are considered as appropriate for selecting the best fitted model in Bayesian approach using BRMS. WAIC is calculated as log predictive density for each data point minus estimated effective number of parameters, and it becomes unreliable if any estimated effective number parameter exceeds 0.4, which holds true for our case. As a result, LOOIC was used to compare the four models fitted and the full model containing both individual level and community level variables was selected as best fitted model because it had the smallest LOOIC than the other three models. Adjusted odds ratio (AOR) with 95% credible interval (CrI) from best fitted model was used to assure variables which have significant association with stunting status among under five children.

## Ethical consideration

Ethical clearance was obtained from the Institutional Review Board of Institute of Public Health, CMHS, and the University of Gondar. Permission for data access was obtained from measure demographic and health surveys through online request from the website http://www.dhsprogram.com

## Results

### Socio demographic and economic characteristics

The median age of children and mothers participated in the study was 31 months (IQR = 17–45) and 29 years (IQR = 24–34) respectively.

Regarding disease status, 16.10%, 22.08% and 22.73% of the children had a history of diarrhea, fever and cough in the last two weeks respectively. Concerning obstetric related factors, 62.70% of the children were delivered at health facility; vast majority, 96.93% of the children were born as single baby and 16.81%, 48.26% and 34.93% of the children were small, average and large in size at birth respectively (Table 1).

### Pooled prevalence of stunting among children aged 6–59 months using recent DHS of 35 SSA

Despite heterogeneity between countries more than one-third 35%, 95%CI: (34.87, 35.31)) of the children in the 35 SSA countries were purportedly stunted. It ranges from the highest in

**Table 1. Socio demographic, economic and Health related and/obstetrical characteristics of children and mothers; from recent DHS data of 35 SSA counties, 2008–2020.**

| Characteristics | Weighted frequency | Proportion (%) |
|---|---|---|
| **Sex of the child** | | |
| Male | 87,387 | 50.37 |
| Female | 86,114 | 49.63 |
| **Place of residence** | | |
| Urban | 53,475 | 30.82 |
| Rural | 120,026 | 69.18 |
| **Wealth index** | | |
| Poorest | 39,194 | 22.59 |
| Poorer | 37,623 | 21.68 |
| Middle | 34,784 | 20.05 |
| Richer | 33,251 | 19.16 |
| Richest | 28,649 | 16.51 |
| **Marital status** | | |
| Married | 153,129 | 88.26 |
| Un married | 20,372 | 11.74 |
| **Sex of household head** | | |
| Male | 138,128 | 79.61 |
| Female | 35,373 | 20.39 |
| **SSA regions** | | |
| Central Africa | 32,127 | 18.52 |
| East Africa | 77,286 | 44.55 |
| South Africa | 3,503 | 2.02 |
| West Africa | 60,585 | 34.92 |
| **Water source** | | |
| Improved | 102,032 | 64 |
| Unimproved | 57,393 | 36 |
| **Type of toilet facility** | | |
| Improved | 61,419 | 38.53 |
| Unimproved | 98,004.70 | 61.47 |
| **Immunization status** | | |
| Complete | 107,564 | 62 |
| Incomplete | 65,937 | 38 |
| **Recent diarrhea** | | |
| No | 145,491 | 83.9 |
| Yes | 27,926 | 16.10 |
| **Fever in last two weeks** | | |
| No | 135,117 | 77.92 |
| Yes | 38,283 | 22.08 |
| **Cough in last two weeks** | | |
| No | 133,962 | 77.27 |
| Yes | 39,397 | 22.73 |
| **Place of delivery** | | |
| Health facility | 108,699 | 62.70 |
| Home | 64,669 | 37.30 |
| **Perceived size at birth** | | |
| Small | 27,772.6 | 16.81 |

*(Continued)*

Table 1. (Continued)

| Characteristics | Weighted frequency | Proportion (%) |
|---|---|---|
| **Sex of the child** | | |
| Average | 79,732.8 | 48.26 |
| Large | 57,718.7 | 34.93 |
| **Types of birth** | | |
| Single | 168,176 | 96.93 |
| Multiple | 5,325 | 3.07 |
| **Deworming in past six months** | | |
| Yes | 70,225 | 43.95 |
| No | 89,562 | 56.05 |
| **Subsequent birth interval** | | |
| <24 | 56,922 | 35.7 |
| ≥24 | 102,502 | 64.3 |

Burundi, 59%, 95%CI: (58.02, 60.59) to the lowest in Gabon, 17, 95%CI: (15.23, 18.16). Since there is heterogeneity between countries ($I^2 > 50\%$), we have conducted subgroup analysis by considering the sub-regions of SSA but still the heterogeny within the sub-groups is concerning. Accordingly, the pooled prevalence of stunting among children was higher in the East Africa sub-region, 37%, (95%, CI: 36.96, 37.06) and lowest in the South Africa sub region 27, (95%CI: 25.37, 28.28) (Fig 1).

## Associated factors of stunting among children aged 6–59 months in SSA

Four models were fitted based on Bayesian statistical approach to identify predictors of stunting status among under five children. Models' convergence to targeted distribution were

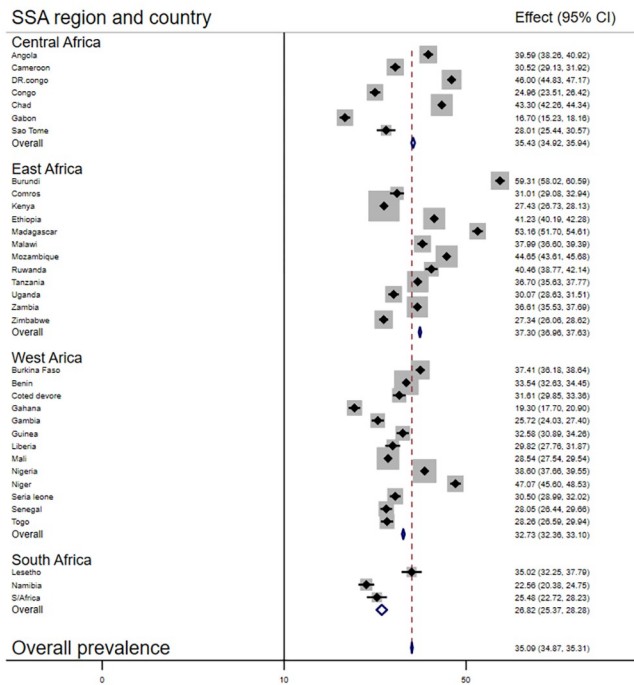

**Fig 1. Forest plot of Pooled prevalence of stunting among children aged 6–59 months: Evidence from recent DHS data of 35 SSA countries.**

assessed using trace plot, density plot, the value of R-hat and adequacy of effective sample size. A well converged model is the one with well-mixed chains in the trace plot, symmetric distribution in the density plot, R-hat value close to one and effective sample size greater than one thousand. In our case, the R hat values for all parameters were one, the effective sample size was adequate (>1000), the chains were well mixed in the trace plot and symmetric distribution attained in the density plot. After assuring the convergence to the targeted distribution, LOOIC was computed to select the best fitted model to the data and a model with smallest LOOIC is considered best fitted model to the data. Based on LOOIC, the final model, the one containing both individual and community level predictors, was selected as the best fitted model.

The ICC in the null model was 7.84%. These indicated that 7.84% of the total variability of stunting status was due to differences between clusters/EA, and the rest unexplained 81.94% was attributable to the individual differences. MOR of the null model was 1.65 which implied that children within a cluster of having higher risk of stunting status had 1.65 times higher chance of having stunting status compared to children within a cluster of having lower risk of stunting status if children were picked randomly from two different clusters (EAs). came to PCV, about 28.57% of the variability in stunting status of children was explained by the full model. The median odds ratio also revealed that stunting of under-five children was heterogeneous among clusters.

We have included both individual and community level factors simultaneously in the final Bayesian hierarchical multi-variable logistic regression analysis.

Of the individual level factors included in the study; child age, sex of the child, type of birth, perceived size of the baby at birth, breastfeed, history of fever in the last two weeks, type of toilet facility, type of drinking water source, place of delivery, history of diarrheal disease in the last two weeks, maternal education, wealth index and maternal mass media exposure were found to be significantly associated with stunting of children. Whereas factors including place of resident and SSA region were significant predictors of stunting status of children at community level.

The Bayesian multi-level multi variable binary logistic regression model showed that a one month increase in child's age was associated with 1% higher odds of stunting. Regarding sex, the odds of getting stunted in male children was 27% (AOR = 1.27, 95% CrI 1.25, 1.30) higher compared to female children. Being a multiple birth increases the odds of getting stunted by 75% (AOR = 1.75%, CrI 1.64, 1.85) compared to a singleton born baby, and perceived size of the baby at birth; being small size at birth was associated with 29% (AOR = 1.29, CrI 1.25, 1.32) increased odds of stunting compared to average birth size children whereas being large size at birth reduces the odds of getting stunting by 14% (AOR = 0.86, CrI 0.84, 0.88) than average birth size baby. Another significant predictor of stunting status of children at individual level were breastfeeding -and history of fever; in which children who had no history of breastfeeding were found to be associated with 10% (AOR = 1.10, CrI,1.03, 1.17) raised odds of stunting than their counterparts, and children who had a history of fever in the last two weeks were associated with 9% (AOR = 1.09, CrI 1.06, 1.12) higher odds of getting stunted than their counterparts.

In addition to individual level predictors illustrated above factors like, type of source of drinking water and types of toilet facility were significantly associated with stunting status of children in which children from household who had not improved water source and toilet facility had 7% (AOR = 1.07, CrI 1.04, 1.09) and 8% (AOR = 1.08, CrI 1.05, 1.11) higher odds stunting compared to those children from household who had improved source of drinking water and improved toilet facility. Regarding place of delivery, children born at home were associated with 17% (AOR = 1.17, CrI 1.14, 1.20) escalated odds of stunting than counterparts.

Maternal educational status and stunting of Children were found to be significantly positively associated; children from mothers with no formal education, primary and secondary education increased the odds of getting stunting by 3.07 times (AOR = 3.07, CrI 2.79, 3.39), 2.62 times (AOR = 2.62, CrI 2.38, 2.88) and 1.86 times (AOR = 1.86, CrI 1.69, 2.04) than children born to mothers with higher educational level.

Wealth index was a significant predictor of stunting as well. The odds of getting stunted was 12% (AOR = 1.12, 95%CrI:1.08, 1.15) and 8% (AOR = 1.08, 95%CrI: 1.04, 1.11) higher among children from poorest and poorer households respectively compared to children from middle wealth household. On the other hand, a child born from richest and richer wealth household had 33% (AOR = 0.67, 95%CrI: 0.64, 0.70) and 8% (AOR = 0.92, 95%CrI: 0.88, 0.95) increased odds of stunting than children of middle wealth household.

Among the community level predictors included in our study, place of residence and region of SSA were significantly associated with stunting of children.

Children belonging to rural households were associated with 16% (AOR = 1.16, CrI 1.12, 1.20) higher odds of stunting compared to those children belonging to urban households.

Considering sub-regions of SSA, being in Eastern Africa sub-region were associated with 7% (AOR = 1.07, CrI 1.04, 1.11) increased odds getting stunting compared to being in central African sub- region but being in western and southern African regions were associated with 21% (AOR = 0.79%, CrI 0.81, 0.97) and 11% (AOR = 0.89, CrI 0.76, 0.82) increased odds of stunting of children respectively (Table 2).

## Discussion

The finding of this study implied that effective policies and strategies that can improve predictors of stunting of children such as women's literacy, household income, health status of mothers and children should be designed and implemented to break the vicious cycle of stunting. Otherwise stunting may remain the economic, health service and psychosocial burden of the SSA countries.

Even though stunting reduction is staggering in sub-Saharan Africa, World Bank's stunting reduction initiative called all hands-on deck stunting reduction through multisectoral efforts in Sub-Saharan Africa which emphasizes engaging additional sectors like Agricultural, education social protection; and water, sanitation, and hygiene (WASH), and UNICEF's stunting reduction effort which focuses on maternal, infant and young child nutrition, and micronutrient supplementation; management of severe acute malnutrition and nutrition in emergencies as well as water and sanitation in communities are promising programmatic efforts being undertaken to speed up stunting reduction in SSA particularly in East Africa.

This Bayesian multi-level binary logistic regression analysis was carried out based on 173, 483 children from 35 SSA countries' recent DHS data 2008 to 2020, and aimed to estimate the pooled prevalence and to identify major contributing factors of stunting among children aged 6–59 months.

Based on our study, the pooled prevalence of stunting among children aged 6–59 months in SSA was about 35% 95%CI (34.87, 35.31). This finding is in agreement with a systematic review and meta-analysis of DHS report 35 low- and middle-income countries worldwide [6].

Our study demonstrated that sex of the child was an important significant predictor of stunting status of a child in which a male child was at higher risk of being stunted than female one. This finding is supported by a meta-analysis of 16 SSA countries and other studies conducted in Zambia and Ethiopia [9–11] The possible justification for this could be male children especially in developing countries like SSA, are more likely to be affected by childhood morbidities particularly acute respiratory tract infections and diarrheal diseases than female

**Table 2. Multivariable binary logistic regression analysis based on hierarchical Bayesian statistical approach result of both individual and community level factors associated with stunting of children: From recent DHS of 35 SSA countries, 2008–2020.**

| Characteristics | Empty model | Individual model AOR (95% CrI) | Community lev. model AOR (95% CrI) | Full model AOR (95% CrI) |
|---|---|---|---|---|
| **Child age** | | 1.01(1.01, 1.02) | | 1.01(1.01, 1.02) |
| **Sex of the child** | | | | |
| Female | | 1 | | 1 |
| Male | | 1.29(1.26, 1.31) | | 1.27(1.25,1.30 |
| **Maternal age** | | | | |
| 21–34 | | 1 | | 1 |
| ≤20 | | 1.12(1.09, 1.18) | | 1.14(1.09, 1.18) |
| ≥35 | | 1.19(1.14, 1.25) | | 1.20(1.15, 1.26) |
| **Media exposure** | | | | |
| Yes | | 1 | | 1 |
| No | | 1.23(1.19, 1.25) | | 1.18(1.15, 1.20) |
| **Wealth index** | | | | |
| Middle | | 1 | | 1 |
| Poorer | | 1.09(1.06, 1.12) | | 1.08(1.04, 1.11) |
| Poorest | | 1.13(1.09, 1.16) | | 1.12(1.08, 1.15) |
| Richer | | 0.88(0.85, 0.91) | | 0.92(0.88, 0.95) |
| Richest | | 0.62(0.60, 0.65) | | 0.67(0.64, 0.70) |
| **Marital status** | | | | |
| Married | | 1 | | 1 |
| Un married | | 1.04(1.01, 1.08) | | 1.02(0.99, 1.07) |
| **Maternal education** | | | | |
| Higher | | 1 | | 1 |
| Primary | | 2.76(2.52, 3.03) | | 2.62(2,38, 2.88) |
| Secondary | | 1.84(1.68, 2.02) | | 1.86(1.69, 2.04) |
| No education | | 2.95(2.68, 3.23) | | 3.07(2.79, 3.39) |
| **Sex of household head** | | | | |
| Female | | 1 | | 1 |
| Male | | 1.02(0.99, 1.05) | | 1.02(0.99, 1.05) |
| **Source of drinking water** | | | | |
| improved | | 1 | 1 | |
| unimproved | | 1.10(1.07, 1.12) | | 1.07(1.04, 1.09) |
| **Type of toilet facility** | | | | |
| improved | | 1 | | 1 |
| Unimproved | | 1.11(1.08, 1.14) | | 1.08(1.05, 1.11) |
| **Breast feed** | | | | |
| Yes | | 1 | | 1 |
| No | | 1.11(1.05, 1.18) | | 1.10(1.03, 1.17) |
| **Recent diarrhea** | | | | |
| No | | 1 | | 1 |
| Yes | | 1.17(1.13, 1.21) | | 1.17(1.14, 1.21) |
| **Taking medication for IP** | | | | |
| No | | 1 | | |
| Yes | | 1.09(0.99, 1.11) | | 1.06(0.98,1.08) |
| **Fever in last two weeks** | | | | |
| No | | 1 | | 1 |

*(Continued)*

**Table 2.** (Continued)

| Characteristics | Empty model | Individual model AOR (95% CrI) | Community lev. model AOR (95% CrI) | Full model AOR (95% CrI) |
|---|---|---|---|---|
| Yes | | 1.08(1.05, 1.11) | | 1.09(1.06. 1.12) |
| **Cough in last two weeks** | | | | |
| No | | 1 | | 1 |
| Yes | | 0.98(0.95, 1.01) | | 0.99(0.96, 1.05) |
| **Place of delivery** | | | | |
| Health facility | | 1 | | 1 |
| Home | | 1.20(1.17, 1.22) | | 1.17(1.14,1.20) |
| **Immunization status** | | | | |
| complete | | 1 | | 1 |
| incomplete | | 1.06(0.98, 1.03) | | 1.03(0.97, 1.01) |
| **Types of birth** | | | | |
| Single | | 1 | | 1 |
| multiple | | 1.75(1.64, 1.85) | | 1.75(1.64, 1.89) |
| **Perceived size at birth** | | | | |
| Average | | 1 | | 1 |
| Small | | 1.29(2.25, 2.32) | | 1.29(1.25, 1.32) |
| Large | | 0.85(0.83, 0.87) | | 0.86(0.84, 0.88) |
| **Birth interval in month** | | | | |
| $\geq$24 | | 1 | | 1 |
| <24 | | 1.15(1.12, 1.18) | | 1.15(1.11, 1.18) |
| **Place of residence** | | | | |
| Urban | | | 1 | 1 |
| Rural | | | 1.96(1.89, 2.00) | 1.16(1.12, 1.20) |
| **SSA regions** | | | | |
| Central Africa | | | 1 | 1 |
| East Africa | | | 0.91(0.87, 0.93) | 1.07(1.04, 1.11) |
| South Africa | | | 0.66(0.61, 0.71) | 0.89(0.81, 0.97) |
| West Africa | | | 0.76(0.74, 0.79) | 0.79(0.76, 0.82) |
| **Distance to health facility** | | | | |
| Not big problem | | | 1 | 1 |
| big problem | | | 1.09(1.06, 1.11) | 0.99(0.97, 1.01) |
| $\delta_u^2$ | 0.28(0.26,0.31) | 0.20(0.19, 0.23) | 0.24(0.22, 0.26) | 0.20(0.18, 0.22) |
| **ICC (%)** | 7.84(5.46,8.19) | 5.7(4.9, 6.7) | 6.8(5.07, 7.21) | 5.70(4.91, 6.58) |
| **PCV (%)** | 0 | 28.57 | 14.29 | 28.57 |
| **MOR** | 1.65(1.62,1.70) | 1.53(1.51, 1.58) | 1.59(1.56, 1.62) | 1.53(1.50, 1.56) |
| **LOOIC** | 225443.1 | 2214563.5 | 221447.0 | 219780.10 |

$\delta_u^2$ = group level variance, AOR = Adjusted Odds Ratio, CrI = credible interval

children even though the biological mechanism is poorly understood [46, 47]. On the contrary, a study done in Ethiopia revealed that female children were associated with increased odds getting stunted than male ones [12].This is probably due to socio-cultural differences in caring for children for example in some societies there is favoring a mild child in nourishing and caring when feel sick than a female child [48, 49].

Based on our study, the household wealth index had a significant positive association with stunting status of children. Children of poorest and poorer households were associated with

higher odds of getting stunted than children of middle wealth households. Whereas children from richer and richest households were associated with lower odds getting stunted compared to children of middle wealth households. This finding is in agreement with a systematic review and meta-analysis in Africa from 2000 to 2013 [26], and other studies conducted in low-income countries such as Nepal, Zambia and Nigeria [20, 50, 51]. This is obvious that richer and richest households are more likely to have food security and diversity [52]. In addition, rich households could have a better health seeking behavior, and may visit a health facility soon when their children feel sick which may have a positive contribution for the health and wellbeing as well as appropriate growth and development of children [53].

In this study, maternal educational level was found to have significant association with stunting status of under five children. As maternal educational level increases from no education to primary, secondary and higher education, stunting status of children decreases. This finding is supported by a systematic review of articles published from 2004 to 2014 worldwide, and other studies in Zambia and Nepal [25, 50, 51, 54].This can be justified by the fact that educated mothers have knowledge about child feeding and its importance for the growth and development of their children [55]. Additionally, maternal education can increase the health knowledge and household income that will have a positive impact on the health and nutritional status of their children [56].

Another variable that has a significant positive association with stunting of children exposed in our study was diarrhea in which, children with a history of diarrhea were associated with an increased odds of getting stunted than children who had not diarrhea. This finding is in agreement with other studies in Nepal, Libya and Ethiopia [19, 21, 22]. The reason might be related to the fact that diarrhea can reduce food intake, loss and malabsorption of nutrients that are vital for growth and development of a child [57].

This study also revealed that fever was a significant predictor of stunting status of children. Children with a history of fever had higher odds of getting stunted than children who did not have fever, and this finding is supported by other studies conducted in Indonesia and Malawi [58, 59]. These could be due to increment of metabolic demand and reduced food intake as a result of the fever [60], and also fever can be a symptom of repeated infections like malaria which may lead to faltering growth of children [61]. Breastfeeding was significantly associated with stunting status of children in this study in which children who had not breastfed were associated with an increased odds of stunting than counterparts. The reason could be breast milk has balanced nutrients that are building blocks of the immune system which can reduce childhood morbidities such as diarrhea and respiratory infections [62], and this was in agreement with studies conducted in Philippines and Indonesia [17, 18].

In this study, unimproved water sources and toilet facilities were positively associated with stunting of children. Children from households who had not access to improved water sources and toilet facilities were found to be more stunted than children from households who had improved water sources and toilet facilities. This study is in line with a systematic review and meta-analysis of 171 DHS of 70 low and middle income countries worldwide from 1984 to 2007 and a study done in Ethiopia [23, 24]. The reason may be due to the fact poor sanitation poor quality of source drinking like polluted with Escherichia coli can affect children's health and nutritional status via a variety of mechanisms such as repeated episode of diarrhea, parasites, environmental enteropathy and other possible ways that restrict nutrient uptake and absorption [63].

Perceived size of the baby at birth had significant association with stunting status of children in which children who had small birth size were more likely to be stunted than large sized children at birth. This finding is supported by other studies conducted in Africa and other developing countries in Indonesia and Ethiopia [10, 14, 58]. Because small birth size may be

manifestation of conditions such as preterm birth, poor maternal nutrition and illness during pregnancy which might cause restricted growth and development of children [64]. Small birth size is also associated with different childhood morbidities like diarrhea and acute febrile illnesses which may lead to stunting of children [65].

Results from this study demonstrated that multiple birth was a significant predictor of stunting of children. A multiple (twin) birth child had an increased odds of becoming stunted than a singleton born child, and this finding is in agreement with other studies conducted in India and Ethiopia [10, 15, 16]. The possible explanation could be that multiple birth is commonly associated with preterm birth as well as perinatal and neonatal morbidities which can interfere with appropriate growth and development of children [66].

In this study, a place of delivery had significant association with stunting of children, that is children born at health facilities were less likely to be stunted than those born at home, and this in agreement with a study conducted in Ethiopia [34]. The possible explanation can be related to the fact that, a child born at health facility and his/her mother is more likely to receive proper postnatal care like vaccination which, is very crucial for the appropriate growth and development of the child as it can prevent several vaccine preventable diseases and supply the child with vital nutrients [67].

Results of this study also demonstrated that media exposure of mothers had significant association with stunting of children in which a child whose mother had mass media exposure was associated with reduced odds of being stunted than counterpart, and this finding is in agreement with other studies conducted in Pakistan and Bangladesh [28, 33]. This is due to the fact that Media has a great role in building health and nutrition-related behaviors, attitudes and practices as well as in promoting sociocultural and economic development that might contribute to improved nutritional outcomes [68].

In this study, community level variables; place of residence and SSA region were significantly associated with stunting of children in which children from rural resident parents had higher odds of getting stunted than counterparts, and his finding is supported by other studies conducted in Pakistan and Ethiopia [29, 30]. This may be because better nutritional status of urban children results from maternal prenatal and birth care, quality of complementary feeding and immunization of children. Moreover, urban children are advantageous in nutritional status over rural children in terms of family employment conditions as well as social and family networks to access health care and other social services [69].

This study showed that children in the East African region were associated with increased odds of stunting compared to the Central African region. Whereas children in the Southern and Western African region were less likely to be stunted than those children in the Central African region. This finding is supported by other systematic review and meta-analysis done in SSA [27]. The possible justification may be Eastern African sub-region is frequently associated famine and starvation resulted from repeated drought, storm of conflict and locust swarm which in turn lead to insufficient food production, availability, intake as well as poor nutritional value, and children are highly vulnerable for such type of catastrophes, and my develop many form of undernutrition including stunting [70].

The findings of this study will enable policy makers and program designers to ascertain the most consistent factors associated with stunting of children in SSA for policy inputs in the effort to reduce stunting of children in the region and each sub-region of SSA. Such policy actions should focus on improving women's literacy, increasing household income, improving health status of mothers through infectious diseases prevention and delivery of quality health care, increase access to safe water supply and promote good hygiene and sanitation practices especially in the rural settings. This study can also enable NOGs and donors to identify and

focus on the most stunted affected sub-regions. Furthermore, this study can serve as a baseline for future studies in SSA and respective sub-regions.

Even though stunting reduction is staggering in sub-Saharan Africa, World Bank's stunting reduction initiative called all hands-on deck stunting reduction through multisectoral efforts in Sub-Saharan Africa which emphasizes engaging additional sectors like Agricultural, education social protection; and water, sanitation, and hygiene (WASH), and UNICEF's stunting reduction effort which focuses on maternal, infant and young child nutrition, and micronutrient supplementation; management of severe acute malnutrition and nutrition in emergencies as well as water and sanitation in communities are promising programmatic efforts being undertaken to speed up stunting reduction in SSA particularly in East Africa.

## Strength and limitation of the study

The study was based on SSA countries large dataset which can increase the power of the study to detect the true effect size and to enhance the generalizability. Moreover, this study was done based on the Bayesian approach which incorporates prior information in addition to the data.

Besides, combined data of 35 SSA countries was used to estimate the pooled prevalence of stunting among children, the heterogeneity between countries was significant and highly concerning. Additionally, some modifiable factors such as anemia level, ANC visit during pregnancy and maternal nutritional status such as maternal BMI and height were not included in the analysis because of no observation in some country's dataset. Health system variables like health insurance system coverage, and number and type of health facilities, and other country level variables like political situation of the countries were not assessed in this study.

## Conclusion

This study revealed that the prevalence of stunting in SSA is high. It is particularly highest in East Africa and Central Africa sub-regions. Among SSA countries Burundi, Madagascar, Niger and Democratic republic Congo shared the highest burden of stunting of children.

Our study demonstrated that both individual and community level variables were significantly associated with stunting of under five children.

At individual level increased child's age, being a male child, being a twine birth, being small sized at birth, a child delivered at home, have not breastfed, having fever, having diarrhea, subsequent birth interval less than 24 months being in poorer and poorest wealth households, no formal, primary and secondary education of mother, being in household that have unimproved water source and unimproved toilet facility, no media exposure of the mother were predictors associated with increased odds of stunting of children.

At community level Rural resident and East Africa sub-region were predictors that had significantly associated getting higher odds of stunting of children. Therefore, governmental and non-governmental organizations design initiatives and programs that can increase the access and affordability of maternal and child health services such as health facility delivery, diagnosis and treatment of diarrheal and febrile illnesses. For the communication sector it is better to increase mass media coverage and advocate child health and nutrition related programs. For researchers we would like to recommend to include other factors such as the underlying medical condition of children and mothers, and use more advanced study designs that enable establishing cause and effect relationships.

## Supporting information

**S1 File.**
(DOCX)

## Acknowledgments

Our thanks go to MEASURE DHS Program their permission to use the SSA DHS data set.

## Author Contributions

**Conceptualization:** Bayley Adane Takele.

**Data curation:** Bayley Adane Takele.

**Formal analysis:** Bayley Adane Takele, Lemma Derseh Gezie, Tesfa Sewunet Alamneh.

**Funding acquisition:** Bayley Adane Takele.

**Investigation:** Bayley Adane Takele, Lemma Derseh Gezie, Tesfa Sewunet Alamneh.

**Methodology:** Bayley Adane Takele, Lemma Derseh Gezie, Tesfa Sewunet Alamneh.

**Project administration:** Bayley Adane Takele, Lemma Derseh Gezie, Tesfa Sewunet Alamneh.

**Resources:** Bayley Adane Takele, Lemma Derseh Gezie, Tesfa Sewunet Alamneh.

**Software:** Bayley Adane Takele, Lemma Derseh Gezie, Tesfa Sewunet Alamneh.

**Supervision:** Bayley Adane Takele, Lemma Derseh Gezie, Tesfa Sewunet Alamneh.

**Validation:** Bayley Adane Takele, Lemma Derseh Gezie, Tesfa Sewunet Alamneh.

**Visualization:** Bayley Adane Takele, Lemma Derseh Gezie, Tesfa Sewunet Alamneh.

**Writing – original draft:** Bayley Adane Takele.

**Writing – review & editing:** Bayley Adane Takele, Lemma Derseh Gezie, Tesfa Sewunet Alamneh.

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
