## [Decision Letter · Decision Letter 0]

17 May 2022

PONE-D-21-28714Pooled Prevalence of Stunting and Associated Factors among Children Aged 6-59 Months in Sub-Saharan Africa Countries: A Bayesian multilevel approachPLOS ONE

Dear Dr. Adane,

Thank you for submitting your manuscript to PLOS ONE. After careful consideration, we feel that it has merit but does not fully meet PLOS ONE’s publication criteria as it currently stands. Therefore, we invite you to submit a revised version of the manuscript that addresses the points raised during the review process.

We look forward to receiving your revised manuscript.

Kind regards,

Orvalho Augusto, MD, MPH

Academic Editor

PLOS ONE

**Journal requirements:**

A clean copy of the edited manuscript (uploaded as the new *manuscript* file).

3. PLOS requires an ORCID iD for the corresponding author in Editorial Manager on papers submitted after December 6th, 2016. Please ensure that you have an ORCID iD and that it is validated in Editorial Manager. To do this, go to ‘Update my Information’ (in the upper left-hand corner of the main menu), and click on the Fetch/Validate link next to the ORCID field. This will take you to the ORCID site and allow you to create a new iD or authenticate a pre-existing iD in Editorial Manager. Please see the following video for instructions on linking an ORCID iD to your Editorial Manager account: https://www.youtube.com/watch?v=_xcclfuvtxQ.

**Additional Editor Comments:**

The authors estimate a common prevalence in sub-Saharan Africa using 35 DHS surveys conducted between 2008 and 2018. As expected they found a very high prevalence across the continent and similar factors and conclusions as previously published analyses have shown. The only new information from this analysis is that there remains a large unexplained variance given their heterogeneity measures (ICC, MOR etc).

We know that stunting is not spread all over the country. There are pockets of it in a country. So, why not a sub-country prevalence pooling? That would help more each country to act. I would suggest the authors to at least discuss this.

Major issues:

1. Introduction/Background

Please cut this current introduction to something close the half of what it is now.

Please ensure a flowing text. Eg: Remove the Bayesian argument in the background (you may put part of it in the methods); avoid orphan statements like “Globally, approximately 155 million children under five suffer from stunting” and the “A cross-sectional study conducted using evidence from thirty-five …”

2. Statistical analysis

In a very confusing way, the authors mention survey sampling analysis with Stata (I guess to compute prevalence which was synthesized into the forest plot in figure 1) and child individual-level analysis with brms/R (for the analysis of factors). This must be clarified:

- For the prevalence pooling please i) put details on how country-level prevalence was estimated, not just throw software name (eg: state at least what kind of variance estimator are using for the proportions here). ii) please add information on how the synthetization happened (inverse variance weighting? Random-effects modelling? Logit transformation? Etc etc… what software was used here? It seems to be Stata).

- For Bayesian analysis and factors study: Reduce the text please. Lots of what is written is “jargon” of Bayesian arguments. You can move a lot of this to supplementary materials. For example: You put a very general prior consideration but no information of the priors used in this analysis!

3. The discussion in the limitations fails to recognize that you do not have health systems, governance and particular high-level country and region variables.

Minor issues:

1. Please cite the software you use:

Stata

R

And brms as an R library/package. This is critical to cite the paper from Paul-Christian Bürkne.

2. Table 1 and 2 could be placed together. And in the supplementary materials we should have it per survey.

3. Figure 1 needs more work.

- Prevalence will never be negative. So why are negative labels there?

Reviewers' comments:

Reviewer's Responses to Questions

**Comments to the Author**

1. Is the manuscript technically sound, and do the data support the conclusions?

Reviewer #1: Yes

2. Has the statistical analysis been performed appropriately and rigorously? 

Reviewer #1: I Don't Know

3. Have the authors made all data underlying the findings in their manuscript fully available?

Reviewer #1: Yes

4. Is the manuscript presented in an intelligible fashion and written in standard English?

Reviewer #1: No

5. Review Comments to the Author

Reviewer #1: Thank you very much for submitting this manuscript. This is one of the important topics to improve child survival in Sub-Saharan Africa. Following are my comments/suggestions:

1. Abstract:

1.1. In methods, kindly give the duration of DHS datasets.

1.2. In the results, there are several grammatic errors, please correct these.

1.3. Your conclusion is not according to the results presented in the results. Kindly correct it.

2. Main manuscript:

2.1. Background: Very long and it is not presented well. Usually, the background should depict epidemiology, what is known, what is unknown and how this research generates evidence to fill the gap in knowledge. kindly rewrite the background.

2.2. Methods: you have stated, .."In 2015, Sub-Saharan Africa was home to 27 of the world’s 28 poorest countries and had more extremely poor people than in the rest of the world combined." Kindly give a reference to support this statement.

2.3. Methods: Individual-level variables - Why breastfeeding was not mentioned here. Please explain whether you used ever breastfed and currently breastfeeding? further, complementary feeding is very important. Why you did not use this in the final model. please consider this variable to improve the current analysis.

2.4. Methods: You stated, "These datasets were appended, cleaned, and recoded according....." Please explain how did you clean the data. Write all the steps of DHS data cleaning once you downloaded the dataset from the public domain.

2.5. Results: Table 1: Please be consistent with decimal points.

2.6. Results: Table 2: Place of delivery; kindly use skilled or skilled birth attendants also.

2.7. Results: Figure 1: Please explain the analysis you have done for Figure 1 in the method section. it seems you used meta-analysis.

2.8. Results: Table 3: Child age: please explain this in the text.

2.9. Results: Table 3: Kindly change the reference value of maternal age to show the risk factor for stunting.

2.10. Results: Table 3: SSA regions - East Africa, in the adjusted community-level model it shows a protective effect while in the full model t is one of the risk factors. Please explain it. Further, kindly evaluate the interaction of this variable with any other variable.

2.11. Discussion: Kindly add implications of these findings.

2.12. Discussion: Kindly give a snapshot of the current programmatic efforts in reducing the stunting in East Africa.

6. PLOS authors have the option to publish the peer review history of their article (what does this mean?). If published, this will include your full peer review and any attached files.

Reviewer #1: **Yes: **Yasir Bin Nisar

---

## [Author Response · Author response to Decision Letter 0]

10 Aug 2022

PLOS ONE 

Point by point response for editors/reviewers comments 

Manuscript title: Pooled Prevalence of Stunting and Associated Factors among Children Aged 6-59 Months in Sub-Saharan Africa Countries: A Bayesian multilevel approach

Manuscript ID: PONE-D-21-28714

Dear editor/reviewer. 

Dear all,

We would like to express our deepest gratitude for this constructive, corrective, and improvable comments, questions and suggestions on this manuscript that would improve the content and scientific merit of the manuscript. We considered each comments and suggestion raised by the editor and reviewer on the manuscript thoroughly. We have performed all of the statistical analyses again and addressed all the editor’s and reviewer’s comments in our revised manuscript (both with track changes and without track changes). Our point-by-point responses for each comments and questions are described in bold referenced by page/line of the resubmission.

Response to Editor’s comment

Additional comments 

Authors’ response: We Thank the editor for this interesting comment and we sense the issue raised. But there are plenty of studies that estimates sub-country level stunting prevalence rather the aim of this study was to determine the burden of stunting at regional and sub-regional level which inturn could enable international, regional and sub-regional organizations, and other stakeholders to cooperate and share good experiences to accelerate the staggering reduction of stunting of children in SSA (See Background section, line 104, page 5).

Major issues: 

1. Introduction/Background

Please cut this current introduction to something close the half of what it is now.

Please ensure a flowing text. Eg: Remove the Bayesian argument in the background (you may put part of it in the methods); avoid orphan statements like “Globally, approximately 155 million children under five suffer from stunting” and the “A cross-sectional study conducted using evidence from thirty-five…”

Authors’ response: Thank you for the comments. We accepted and modified it accordingly. (See Background section, page 4-5 ).

2. Statistical analysis

In a very confusing way, the authors mention survey sampling analysis with Stata (I guess to compute prevalence which was synthesized into the forest plot in figure 1) and child individual-level analysis with brms/R (for the analysis of factors). This must be clarified:

- For the prevalence pooling please i) put details on how country-level prevalence was estimated, not just throw software name (eg: state at least what kind of variance estimator are using for the proportions here). ii) please add information on how the synthetization happened (inverse variance weighting? Random-effects modelling? Logit transformation? Etc etc… what software was used here? It seems to be Stata).

-For Bayesian analysis and factors study: Reduce the text please. Lots of what is written is “jargon” of Bayesian arguments. You can move a lot of this to supplementary materials. For example: You put a very general prior consideration but no information of the priors used in this analysis!

Authors’ response: Thank you for the concerns. We accepted and modified it. (see statistical analysis section, page 9-11, and supplementary materials).

3. The discussion in the limitations fails to recognize that you do not have health systems, governance and particular high-level country and region variables.

Authors’ response: Thank you for the comment. We accepted and included it. (see strength and limitation section of the discussion, line 492, page 22).

Minor issues:

1. Please site the software you use; Stata, R and brms as an R library/package. This is critical to cite the paper from Paul-Christian Bürkne.

Authors’ response: Thank you for the concerns. We modified it accordingly. (see data management and analysis of the methods section, line 175, page 8).

2. Table 1 and 2 could be placed together. And in the supplementary materials we should have it per survey

Authors’ response: Thank you for the comment. We accepted and modified it accordingly. (see Table 1, page 29-30, and the supplementary materials).

3. Figure 1 needs more work.

 Prevalence will never be negative. So why are negative labels there?

Authors’ response: Thank you for the careful notice of the figure and identifying the issue. We accepted and modified it accordingly. (see Figure 1).

Response to reviewers comments

1. Abstract

 1.1 In methods, kindly give the duration of DHS datasets.

Authors’ response: Thank you for the comments. We accepted and modified it. (See Abstract section, line 44, page 2).

1.2. In the results, there are several grammatic errors, please correct these.

Authors’ response: Thank you for the concerns. We corrected it in the revised manuscript. (See Result section of the abstract, line 50, page 3).

1.3. Your conclusion is not according to the results presented in the results. Kindly correct it.

Authors’ response: Thank you for the comments. We corrected it in the revised manuscript (see Conclusion part of the abstract, line 56, page 3). 

2. Main manuscript:

2.1 Background:Very long and it is not presented well. Usually, the background should depict epidemiology, what is known, what is unknown and how this research generates evidence to fill the gap in knowledge. kindly rewrite the background.

Authors’ response: Thank you for the comments . We accepted and addressed it in the revised manuscript (see Background section, line 62, page 4). 

2.2. Methods: you have stated, .."In 2015, Sub-Saharan Africa was home to 27 of the world’s 28 poorest countries and had more extremely poor people than in the rest of the world combined." Kindly give a reference to support this statement.

Authors’ response: Thank you for the comments . We addressed it in the revised manuscript (see Methods section, line, 119, page 6). 

2.3. Methods: Individual-level variables - Why breastfeeding was not mentioned here. Please explain whether you used ever breastfed and currently breastfeeding? further, complementary feeding is very important. Why you did not use this in the final model. please consider this variable to improve the current analysis.

Authors’ response: Thank you for the concerns. Dear reviewers, regarding breast feeding we addressed it in the revised manuscript (see Methods section, line 161, page 8). Regarding complementary feeding, it was one of our candidate explanatory variables but we had dropped it because of its high multicollinearity (VIF>10). 

2.4. Methods: You stated, "These datasets were appended, cleaned, and recoded according....." Please explain how did you clean the data. Write all the steps of DHS data cleaning once you downloaded the dataset from the public domain.

Authors’ response: Thank you for the comments . We addressed it in the revised manuscript (see Methods section, line 162 page 8). 

2.5. Results: Table 1: Please be consistent with decimal points. 

Authors’ response: Thank you for the observations. We corrected it in the revised manuscript (see Results section Table 1, page 29-30). 

2.6. Results: Table 2: Place of delivery; kindly use skilled or skilled birth attendants also.

Authors’ response: Thank you for the concern. Skill birth attendants was one of our candidate explanatory variables but we had dropped it from the final model due to its high multicollinearity (VIF>10).

2.7. Results: Figure 1: Please explain the analysis you have done for Figure 1 in the method section. it seems you used meta-analysis.

Authors’ response: Thank you for the comment . We accepted and corrected it in the revised manuscript (see Methods section, 174, page 8). 

2.8. Results: Table 3: Child age: please explain this in the text.

Authors’ response: Thank you for the concern. We explained it accordingly (see Results section, line 307, page 14).

2.9. Results: Table 3: Kindly change the reference value of maternal age to show the risk factor for stunting.

Authors’ response: Thank you for the comment. We modified it (see Results section, Table 2, page 32).

2.10. Results: Table 3: SSA regions - East Africa, in the adjusted community-level model it shows a protective effect while in the full model t is one of the risk factors. Please explain it. Further, kindly evaluate the interaction of this variable with any other variable.

Authors’ response: Thank you for the concerns. Dear reviewer as you know it very well in multilevel analysis of hierarchical data the interaction between level two and level variables, level two and other level two variables and between level variables are expected, and we authors checked suspected variables for that, and there was no significant interaction effect that is why we have not reported in the manuscript. 

2.11. Discussion: Kindly add implications of these findings

Authors’ response: Thank you for the comment. We added it (see discussion section, line 476, page 22).

2.12. Discussion: Kindly give a snapshot of the current programmatic efforts in reducing the stunting in East Africa.

Authors’ response: Thank you for the comment. We accepted and included it (see discussion section, line 467, page 21).

We thank again all reviewers for their helpful critiques that have strengthened this research report!

---

## [Decision Letter · Decision Letter 1]

26 Sep 2022

Pooled Prevalence of Stunting and Associated Factors among Children Aged 6-59 Months in Sub-Saharan Africa Countries: A Bayesian multilevel approach

PONE-D-21-28714R1

Dear Dr. Takele,

We’re pleased to inform you that your manuscript has been judged scientifically suitable for publication and will be formally accepted for publication once it meets all outstanding technical requirements.

Kind regards,

Orvalho Augusto, MD, MPH

Academic Editor

PLOS ONE

Additional Editor Comments (optional):

Reviewers' comments:

Reviewer's Responses to Questions

**Comments to the Author**

1. If the authors have adequately addressed your comments raised in a previous round of review and you feel that this manuscript is now acceptable for publication, you may indicate that here to bypass the “Comments to the Author” section, enter your conflict of interest statement in the “Confidential to Editor” section, and submit your "Accept" recommendation.

Reviewer #1: All comments have been addressed

2. Is the manuscript technically sound, and do the data support the conclusions?

Reviewer #1: Yes

3. Has the statistical analysis been performed appropriately and rigorously? 

Reviewer #1: Yes

4. Have the authors made all data underlying the findings in their manuscript fully available?

Reviewer #1: Yes

5. Is the manuscript presented in an intelligible fashion and written in standard English?

Reviewer #1: Yes

6. Review Comments to the Author

Reviewer #1: (No Response)

7. PLOS authors have the option to publish the peer review history of their article (what does this mean?). If published, this will include your full peer review and any attached files.

Reviewer #1: **Yes: **Yasir Bin Nisar

---

## [Editor Report · Acceptance letter]

3 Oct 2022

PONE-D-21-28714R1 

Pooled Prevalence of Stunting and Associated Factors among Children Aged 6-59 Months in Sub-Saharan Africa Countries: A Bayesian multilevel approach 

Dear Dr. Takele:

I'm pleased to inform you that your manuscript has been deemed suitable for publication in PLOS ONE. Congratulations! Your manuscript is now with our production department. 

Kind regards, 

on behalf of

Dr. Orvalho Augusto 

Academic Editor

PLOS ONE